# Intra-individual methylomics detects the impact of early-life adversity

Shan Jiang[1], Noriko Kamei[2], Jessica L Bolton[2], Xinyi Ma[1], Hal S Stern[3], Tallie Z Baram[2,4], Ali Mortazavi[1]

Genetic and environmental factors interact during sensitive periods early in life to influence mental health and disease via epigenetic processes such as DNA methylation. However, it is not known if DNA methylation changes outside the brain provide an "epigenetic signature" of early-life experiences. Here, we used a novel intra-individual approach by testing DNA methylation from buccal cells of individual rats before and immediately after exposure to one week of typical or adverse life experience. We find that whereas inter-individual changes in DNA methylation reflect the effect of age, DNA methylation changes within paired DNA samples from the same individual reflect the impact of diverse neonatal experiences. Genes coding for critical cellular metabolic enzymes, ion channels, and receptors were more methylated in pups exposed to the adverse environment, predictive of their repression. In contrast, the adverse experience was associated with less methylation on genes involved in pathways of death and inflammation as well as cell-fate–related transcription factors, indicating their potential up-regulation. Thus, intra-individual methylome signatures indicate large-scale transcription-driven alterations of cellular fate, growth, and function.

## Introduction

Experience, particularly during sensitive periods early in life, leaves indelible marks on an individual's ability to cope with life's challenges, influencing resilience or vulnerability to emotional disorders (1, 2, 3, 4, 5). There is evidence that the mechanisms by which early-life experiences influence the function of neurons and neuronal networks involve modification of the repertoire and levels of gene expression via epigenetic processes (1, 2, 3, 4, 6, 7, 8, 9, 10, 11). Among epigenetic processes, changes in DNA methylation of individual genes and at the genomic scale have been reported, and these generally correlate with gene expression (2, 6, 12, 13, 14). However, it is not known if DNA methylation changes might provide a useful "epigenetic signature" of early-life experiences in

an individual child. Such a readily accessible measure might serve as a biomarker for vulnerability or resilience to mental illness. Obviously, it is not possible to repeatedly sample DNA from brain cells in humans to assess DNA methylation changes for predicting and preventing disease. Therefore, current approaches use peripheral cells, including white blood cells (WBCs) or buccal swabs (mixed epithelial/WBC), which are available repeatedly and non-invasively. Here, we tested the feasibility of using peripheral DNA samples to assess the impact of diverse neonatal experiences on an individual by directly comparing two samples collected at different time points from the same individual rat in groups exposed to distinct early-life experiences with defined onset and duration. We have previously established that these diverse experiences provoke specific phenotypic outcomes later in life (4, 15, 16). Specifically, we imposed "simulated poverty" by raising pups for a week (from postnatal day P2 to P10) in cages with limited bedding and nesting (LBN) materials. This manipulation disrupts the care provided by the rat dam to her pups and results in profound yet transient stress in the pups, devoid of major weight loss or physical changes. This transient experience provokes significant and life-long deficits in memory and generates increases in emotional measures of anhedonia and depression (15, 16, 17).

Here, we tested if adversity during a defined sensitive developmental period in rats leads to a detectable epigenomic signature in DNA from buccal swab cells. We obtained intra-individual epigenomic signatures of early-life adversity using reduced representation bisulfite sequencing (RRBS) (18) to identify changes in DNA methylation profiles. Comparisons were made both between two samples from an individual rat (P2 versus P10) and between samples from rats subjected to the two neonatal experiences. We found that assessing the overall methylation profile of samples enabled detection of age and development effects (17, 19), distinguishing P2 samples from those obtained on P10 but did not separate the two groups of pups based on their experience. In contrast, the changes in DNA methylation in two samples obtained from the same rat enabled clear differentiation of the control versus the adverse experience, likely by obviating large inter-individual variance. Thus, our findings establish the feasibility of

[1]Department of Developmental and Cell Biology, University of California, Irvine, CA, USA   [2]Department of Pediatrics and Anatomy/Neurobiology, University of California, Irvine, CA, USA   [3]Department of Statistics, University of California, Irvine, CA, USA   [4]Department of Neurology, University of California, Irvine, CA, USA

Correspondence: tallie@uci.edu; ali.mortazavi@uci.edu
Shan Jiang and Noriko Kamei are co-first authors

identifying markers of adverse experiences that portend risk or resilience to mental illness, with major potential translational impact.

# Results

## Methylation level changes across individuals reflect postnatal age rather than early-life experiences

We obtained a mix of epithelial and WBC DNA from individual rat pups twice, on P2 and P10, using buccal swabs (see the Materials and Methods section). We obtained buccal swabs rather than peripheral WBCs for three reasons. First, the swab, lasting seconds, is much less stressful than a painful needle prick to obtain peripheral blood, and this stress might influence methylation in itself. Second, this approach provides a more direct comparison with human studies where ethical reasons preclude needle pain, although buccal swabs are routinely implemented (20, 21). Finally, several studies found that DNA methylation profiles in buccal swab cells are more similar to patterns from several brain regions than methylation profiles in WBCs (21, 22, 23, 24). Following the initial samples collected on P2 from a group of naive pups, the rats were divided into two groups: one was exposed to simulated poverty. The other was reared in a typical environment for one week. Samples from individuals in both groups were collected again on P10. We examined for intra-individual epigenomic signatures of early-life adversity and compared both P10 samples from groups with two divergent neonatal experiences as well as the changes in methylation levels between matched samples from the same individual rat (P2 versus P10; Fig 1A).

DNA methylation status was assessed using RRBS, with libraries sequenced to an average of 20 million mapped reads, and we reliably detected an average of 482,000 CpGs in both samples of the same individual (Fig S1; see the Materials and Methods section). We performed differential methylation analysis between P2 and P10 for each individual and identified 3,417 significantly differential methylation regions (DMRs) after coalescing CpGs within 100 base pairs that were shared in at least two individuals from each experience group (Fig 1B).

We analyzed the DNA methylation levels of these DMRs in P2 and P10 for both the control and adversity-experiencing (LBN) groups across individuals using k-means clustering and observed substantial changes in the DNA methylation level during the one-week interval in both control and LBN (Fig 2A). The DNA methylation levels of individual samples clearly distinguished rats at different ages (Fig 2A). We further performed principal component analysis (PCA) on the percentage of DNA methylation of these DMRs and found that individual samples were separated by age using the first three principal components (up to 62.1% variances explained), indicating a substantial change in DNA methylation associated with age (Figs 2B, S2, and S3). The separation by age still held when cohort effects were considered (Figs S4, S5, and S6). These data demonstrate that development and age modify the buccal swab methylome (19, 23, 25, 26). PC2, which accounts for 20.7% of the variance, was the dominant component distinguishing samples of different ages (Fig 2B). We found that the PC2 DMRs with most positive weights for predicting the increased age (P10) had reduced methylation level in P10, whereas DMRs with most negative PC2 weights had increased methylation level in P10 (Figs S2 and S3). However, the PCA analyses of the P2 and P10 methylome profiles did not separate the control group from the adversity-experiencing group (Fig 2C). Thus, although the level of DNA

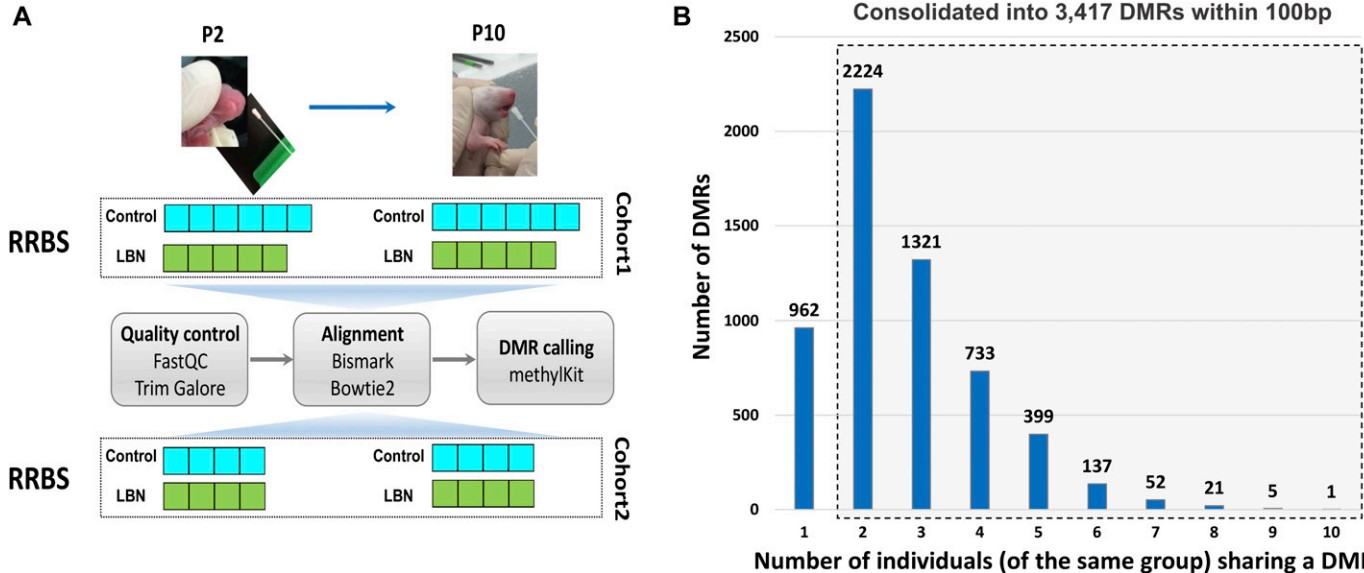

**Figure 1. Experimental design and DMRs calling across individuals.**
**(A)** Experimental design and analysis pipeline. Six control and five LBN individuals were collected for cohort 1; four control and four LBN individuals were collected for cohort 2. **(B)** Histogram of the number of significant DMRs based on the number of individuals sharing the same experience. LBN: a paradigm of simulated poverty and early-life adversity. P2, P10 = postnatal days 2, 10.

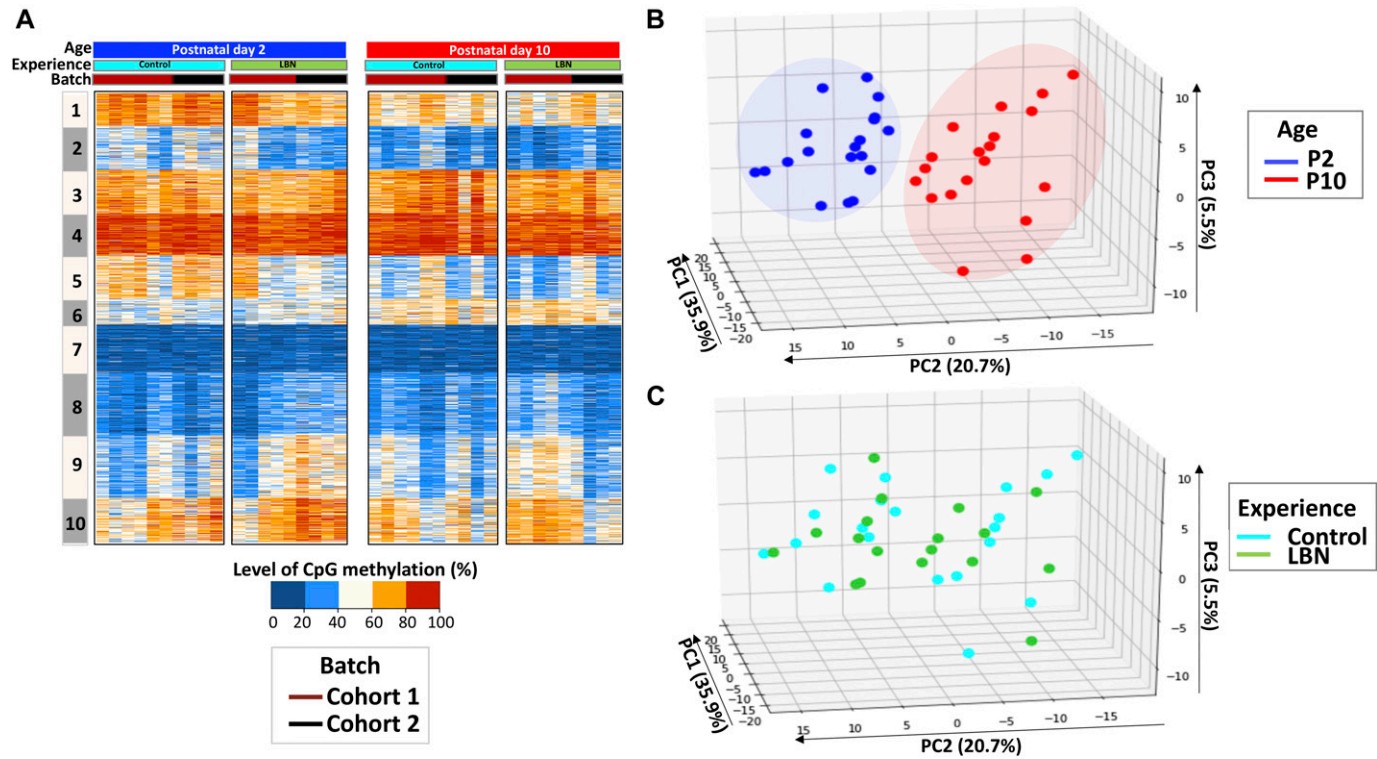

**Figure 2. Separation of individuals by age by profiles of methylation levels on significantly DMRs.**
**(A)** Heat map of CpG methylation percentage on the 3,417 DMRs identified in Fig 1 and showing individual samples. The profile is presented as 10 clusters that are identified using k-means clustering. Blue: low methylation percentage, orange: high methylation percentage. **(B)** PCA of methylation profiles of individual samples, focusing on the 3,417 significant DMRs. Individual samples are labeled by age; blue: P2, red: P10. **(C)** PCA of individual samples focusing on the same DMRs. Individual samples are labeled by experience; cyan: control, green: LBN.

methylation in buccal swabs reflects an epigenetic signature of age, it provides little information about antecedent life experiences.

### Intra-individual changes in methylation can distinguish early-life experience

To probe the impact of the early-life adversity experienced by an individual on DNA methylation patterns of the same individual, we explored intra-individual fold changes in methylation (referred to as "delta methylation," defined as $\log_2(P10/P2)$ of the methylation level of P2 and P10 from the same individual) rather than the absolute value of methylation levels for each pup by taking advantage of the two samples collected immediately before and after a week of imposed adversity. We clustered and aligned these $\delta$ methylation profiles in both early-life experiences. We then examined the intra-individual methylation changes in detail and found that the patterns of changes in methylation within an individual were distinct depending on group assignment (Fig S7). PCA on $\delta$ methylation changes of individual samples reveals that $\delta$ methylation within an individual distinguished the control and LBN groups (Figs 3A and S8). Specifically, the fourth principal component (PC4), accounting for 4.2% of the variances, distinguished most LBNs from controls (Figs 3A and S8). To examine the basis of the separation between LBNs and controls by PC4, we examined the relative contribution of individual DMRs to the overall difference in PC4, and, guided by the slope of the weight distribution, selected a

cutoff threshold at $\pm 2.5 \times 10^{-2}$ to identify 193 DMRs with the most positive weights and 225 DMRs with the most negative weights (Fig 3B and C). Importantly, the adverse and control experiences differentially changed levels of methylation in an experience-specific manner. Thus, within the top-predicting DMRs, the prediction of belonging to the LBN group (afforded by the intra-individual methylation changes in 193 most positive weight–associated DMRs) involved relatively more methylation, compared with controls (Fig 3B). In contrast, intra-individual changes in the 225 most negative weight–associated DMRs suggested generally less methylation level in LBNs than the ones in control experience (Fig 3C). These results indicate that intra-individual changes in methylation-level profiles before and after a defined experience provide a novel epigenetic signature that identifies the nature of the experience.

### Downstream significance of differential methylation resulting from age and experience

The paragraphs above demonstrate that profiles of absolute levels of DNA methylation in mixed epithelial/WBC samples from buccal swab can separate pups by age, whereas the nature of methylation changes in the same individual ($\delta$ methylation) distinguishes different early-life experiences. Although the relations of levels of methylation and of gene expression are not linear, we sought to examine the genes involved in methylation changes related to age and those related to experience.

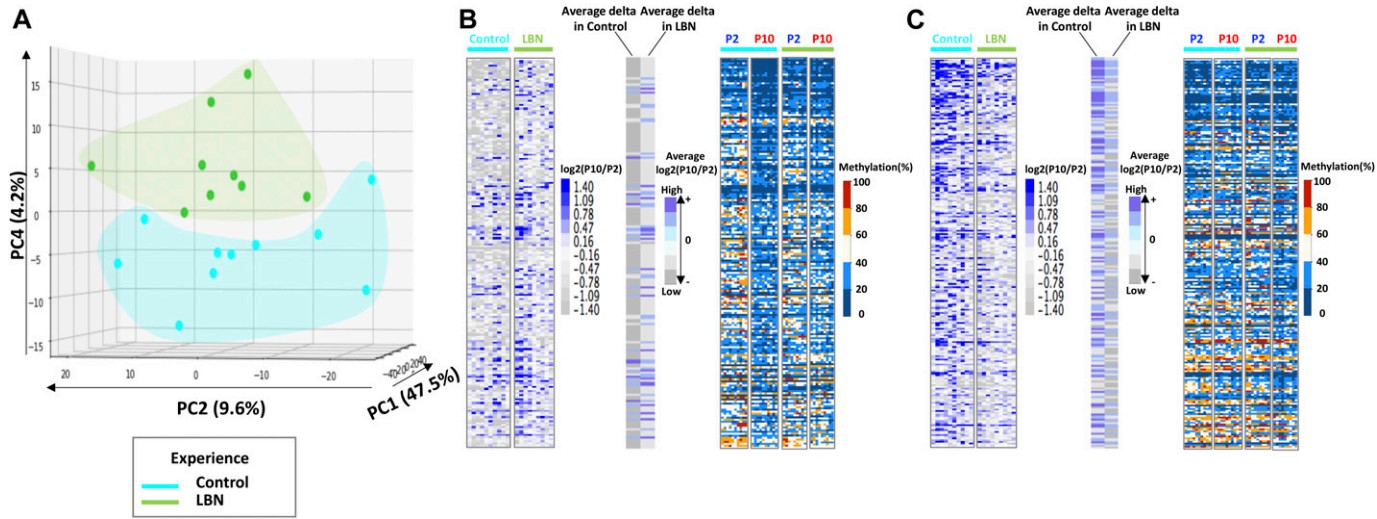

**Figure 3. Intra-individual methylation analysis of significant DMRs separates individual rats by experience.**
**(A)** PCA performed on the difference in methylation levels (δ methylation) between P10 and P2 (log2(P10/P2)) of an individual pup. We focus on the 3,417 identified DMRs and label individuals by experience; cyan: control, green: LBN. PC4 provides the discrimination. **(B)** δ Methylation profile, average δ methylation, and absolute methylation levels (in percent) of 193 DMRs with most positive weights in PC4. DMRs are ranked by weights from high to low. **(C)** δ Methylation profile, average δ methylation, and absolute methylation levels of 225 DMRs with most negative weights in PC4. DMRs are ranked by weights from high to low.

Principal component (PC2) distinguished rats by age (Fig 2B). We focused only on analyzing the control group to obviate potential effect of the adversity experience (Fig S3). We found that 249 DMRs contributed to most of the overall differences in PC2, which predicted the age being P10 in controls. Of these, 135 DMRs were less methylated, whereas 114 DMRs were more methylated in P10 (Fig 4A and Table S1). We performed gene association analysis on these top-predicting DMRs of P10 and found that our 135 most positive weight PC2 DMRs were associated with 105 genes, whereas the 114 most negative weight PC2 DMRs were associated with 91 genes. These age-related genes could be clustered into distinct functional categories (Fig 4B). In general, genes with decreased methylation level, predictive of augmented gene expression, were involved in energy metabolism (Man1c1, B4galt4, Mcart1, Mrc2, Ampd3, and Arhgef17), cytoskeleton and trafficking (Fry, Krt42, RGD130731, Itga6, and Fbxo9), receptors and ion channels (Htr2a, Scarf2, Kcnip1, and Traf3), and cellular responses to growth hormones (Fgfr3, Ltbp1, and Net1). Gene ontology analysis identified gene clusters involved in response to injury, regulation of growth, and ion transport (Fig S3). By contrast, genes with increased methylation (i.e., expected to be less expressed with increasing age) were enriched in transcription (Otx1, Pax9, Dlx4, Irx4, Satb2, and Nr2f2) and kinases (Srcin1, Map3k6, Atp8b4, and Jak3) (Fig 4B). These findings suggest that age-related methylation changes are strongly involved in developmental processes in the neonatal organism.

To characterize the genes influenced by the adverse LBN experience, we performed gene association analysis on the top-predicting PC4 DMRs of LBN (Fig 3) and found that the 193 most positive weight PC4 DMRs were associated with 135 genes, whereas the 225 most negative weight PC4 DMRs were associated with 165 genes (Fig 4C and Table S2). The 193 most positive weight DMRs had generally more methylation compared with controls, which suggests reduced gene expression (relative repression) after the LBN

experience. The corresponding 135 genes coded for critical cellular enzymes and interacting proteins essential for normal metabolism and growth such as cellular cytoskeleton and trafficking (Sys1, Map3k8, and Plekhg5) and cellular metabolism (Mrpl23 and Hs6st1). Other genes within this group include Sipa1, Eif3k, Ttll5, Mark2, Ralgapa2, Net1, H6pd, Tet3, Dapp1, Sulf2, Ppp1r21, Dusp7, and Nudt9. In addition, these PC4 positive weight–associated genes coded for receptors/ion channels and transmembrane-signaling proteins (Chrna9, Grik5, Gpr39, Nrp2, Fzd5, Pcnx1, and Cd83), response to inflammation (Pdcd6ip, Tnfrsf1b, Card10, Traf3, and Cxcr4), and transcription factors responding to growth factors (Sim2, Meis1, Lrrfip, and Rai1) (Fig 4D). The combined expected repression of these genes would lead to disruption of typical growth, metabolism, and maturation processes that are fundamental to the developing organism. In contrast, the 165 genes with the most negative PC4 weight DMRs were generally less methylated in the adversity-experiencing rats (i.e., predicted to be expressed at relatively higher levels) and were strongly enriched in homeobox genes involved in very early cell specification (Six2, Hoxb5, Satb2, Six1, Dlx1, and Nkx2-3), as well as other transcription factors and corepressors (Tfap2c, Skor1, Tbx3, Gata3, Tbx4, and Hr). In addition, the group included genes involved in apoptosis and inflammation, including Dapk1, Gdnf, Mog, and Tnfaip2 (Fig 4D). Increased expression of these genes would indicate a reversion to earlier, more primitive, cell state and evidence of inflammation and reprogramming, perhaps to avoid death.

# Discussion

We find here that comparing cohort-wide DNA samples obtained at different developmental ages reveals the signature of age and development on the peripheral methylome, as widely reported.

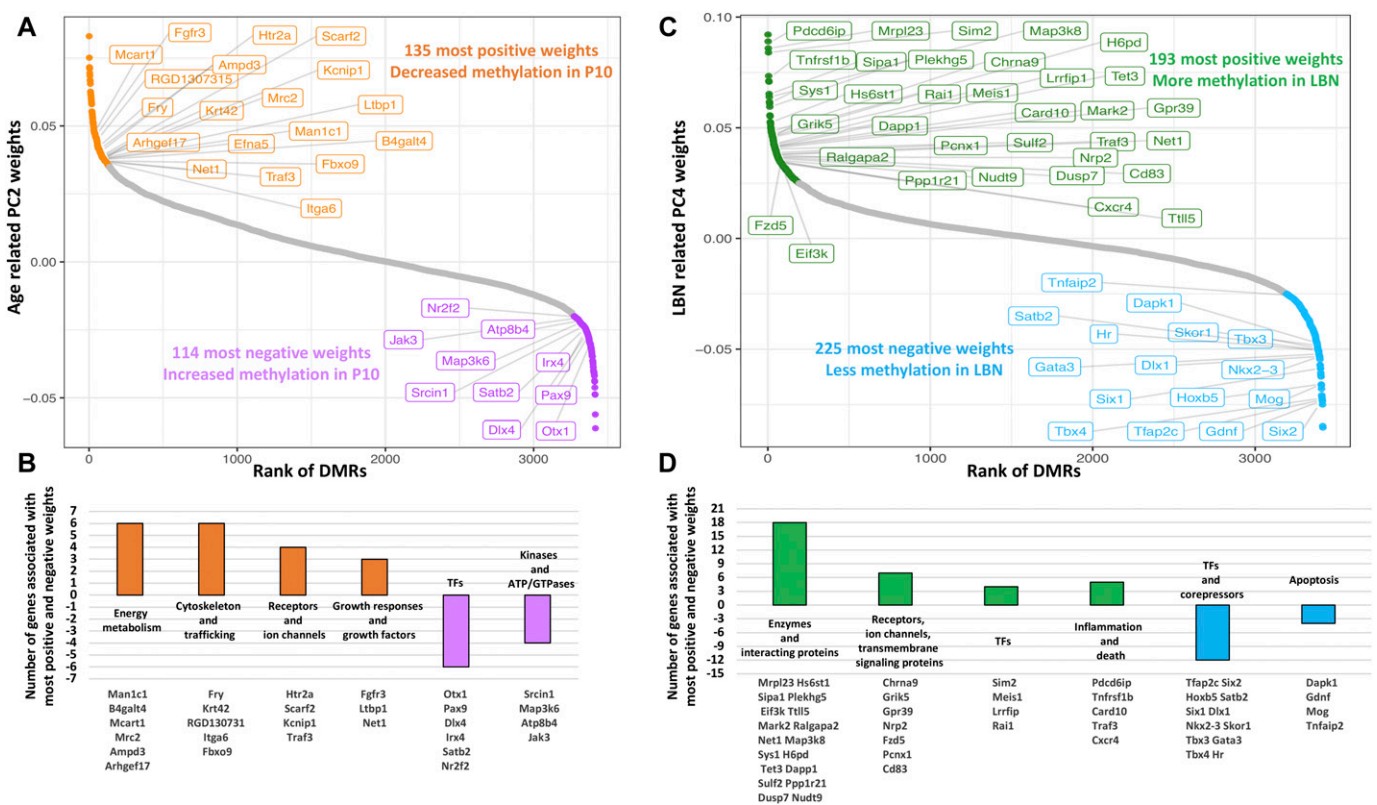

**Figure 4. Expected consequences of age- and experience-related DMRs.**
**(A)** Analysis of the PC2 weights that separate the P2 and P10 samples by age. **(B)** Most positive (orange) and negative (purple) weights are enriched in genes associated with functional categories listed in (B). **(C)** Analysis of PC4 weights that separate individual rats that had experienced early-life adversity (LBN) or typical rearing conditions (controls). **(D)** Most positive (dark green) and negative (sky blue) weights are enriched in genes associated with functional categories listed in (D).

However, these inter-individual analyses do not distinguish the divergent impacts of diverse experiences that take place during the intervening developmental epoch. By contrast, paired samples from the same individual before and after an adverse or typical developmental experience enable clear distinction of each of these experiences: we identify epigenetic "scars" and "kisses" that, at least in the rodent, precede and predict later-life emotional functions.

Although it is known that early-life experiences drive gene expression changes and thus further influence the maturation of brain and other organs in mammalian individuals, our knowledge about specific epigenetic regulations involved into these processes are limited. Among epigenetic regulations, DNA methylation is known to correlate with gene expression changes. However, it is not known if DNA methylation changes might provide a useful "epigenetic signature" of early-life experiences in an individual child. Therefore, this study addresses two critical questions to understand the nature of DNA methylation changes in early-life experiences: (1) Does a short period of early postnatal life change methylation patterns in individuals? (2) Can methylation changes be used to distinguish individuals who had experienced early-life adversity? Consistent with previous studies, we find that simple comparison of methylation levels across a cohort cannot distinguish rats with different early-life experiences, although the signature of growth/age is apparent. We further develop a novel

approach and demonstrate for the first time that intra-individual changes in methylation patterns can robustly distinguish individuals with adverse experiences from those reared in typical conditions, thus potentially serving as a predictive signature in individuals.

Although cognizant of the complex relation of DNA methylation levels and gene expression, we speculate here on the downstream consequences of the potential expression changes of gene families and individual genes that differentiate adverse and typical development. Among the genes differentially methylated in the groups of rats studied here, many overlapped in the LBN and control groups, suggesting that they are modified primarily by age rather than experience. Importantly, our PCA analyses of intra-individual methylation changes identified the PC4 genes that were differentially methylated in the P10 LBN rats compared with the same rats on P2, but that were not affected in P10 versus P2 controls. These genes might then provide information about the processes associated with the early-life adversity experience that might carry long-term consequences.

Indeed, analyses of the top contributing genes to the distinction of having survived adversity in P10 rats was revealing: in LBN rats, there was a striking enrichment of increased methylation (indicative of reduced expression) in genes carrying out typical processes of metabolism, trafficking, and growth. In contrast, there was an expected overexpression (reduced methylation) of gene

families associated with inflammation, death, and reversion to more primitive developmental states. These seem to be orchestrated by differentially methylated transcription factors. How the adversity experience provokes these changes is unclear and may involve molecular signals, including hormones and nutrients that modulate the complex enzymatic processes that govern DNA methylation status (27, 28, 29).

In summary, we show here the influence of a short epoch of adversity during a developmental sensitive period on intra-individual rodent methylome. In future studies, it would be exciting if this DNA methylation signature of early-life adversity is applied in human neonates and infants.

# Materials and Methods

### Animals

Subjects were born to primiparous Sprague Dawley rat dams (around P75) that were maintained in the quiet animal facility room on a 12-h light/dark cycle with ad libitum access to laboratory chow and water. Parturition was checked daily, and the day of birth was considered postnatal day 0 (P0). Litter size was adjusted 12 per dam on P1, if needed. On P2, pups from several litters were gathered, and 12 pups (6 males and 6 females) were assigned randomly to each dam to obviate the potential confounding effects of genetic variables and litter size. Each pup was identified by a rapid (<2 min) foot pad tattooing using animal tattoo ink (Ketchum).

### Early-life adversity paradigm

The experimental paradigm involved rearing pups and dams in "impoverished" cages for a week (P2–P9) as described elsewhere (30, 31, 32). Briefly, routine rat cages were fitted with a plastic-coated aluminum mesh platform sitting ~2.5 cm above the cage floor (allowing collection of droppings). Bedding was reduced to only cover cage floor sparsely, and one-half of a single paper towel was provided for nesting material, creating a LBN cage. Control dams and their litters resided in standard bedded cages, containing 0.33 cubic feet of cob bedding, which was also used for nest building. For each experiment, pups form several litters were mixed and then assigned randomly to a control or an LBN dam. This procedure minimizes the potential effects of pup genetic background on outcomes. Control and experimental cages were undisturbed during P2–P9 and housed in a quiet room with constant temperature and a strong laminar airflow, preventing ammonia accumulation. For technical reasons, the study was conducted in two "batches" (cohorts). These cohorts differed solely in the dates at which they were conducted.

### Collection of buccal swab from each pup

The first buccal swab was collected from both cheeks of each pup before randomization on P2, using a HydraFlock swab (Puritan diagnostics, LLC). After an hour's rest with their mother, a second buccal swab was collected, enabling sufficient DNA from each pup.

Pups were then randomized to controls or LBN cages. During P3–P9, behaviors of dams in both control and adversity/LBN cages was observed daily, to ascertain the generation of fragmented unpredictable caring patterns by the adverse environment (33, 34). On P10, buccal swabs were collected as described for P2, and then all litters were transferred to normal bedded cages.

### Isolation and quantification of DNA for making RRBS libraries from rat buccal swab

The buccal swab was placed into DNA shields (Zymo Research) immediately after swabbing. DNA was prepared from the DNA shields solution using the Quick-gDNA MiniPrep kit (Zymo Research) following the manufacturer's protocol. The quantity of double-stranded DNA was analyzed using Qubit, and RRBS libraries were prepared from 40 ng of genomic DNA digested with Msp I and then extracted with DNA Clean & Concentrator-5 kit (Zymo Research). Fragments were ligated to pre-annealed adapters containing 5′-methyl-cytosine instead of cytosine according to Illumina's specified guidelines (www.illumina.com). Adaptor-ligated fragments were then bisulfite-treated using the EZ DNA Methylation-Lightning kit (Zymo Research). Preparative-scale PCR was performed, and the resulting products were purified with DNA Clean & Concentrator for sequencing. Amplified RRBS libraries were quantified and qualified by Qubit, Bioanalyzer (Agilent), and Kapa Library Quant (Kapa systems) and then sequenced on the Illumina NextSeq 500 platform.

### RRBS data processing and detection of DMRs

Adaptor and low-quality reads were trimmed and filtered using Trim Galore! 0.4.3 (35) with the parameter "--fastqc –stringency 5 –rrbs –length 30 –non_directional." Reads were aligned to the rat genome (RGSC 6.0/rn6) by using Bismark 0.16.3 (36) with "---non_directional" mode. CpG sites were called by "bismark_methylation_extractor" function from Bismark. Single CpG sites with more than 10-reads coverage were kept for DMR calling. Differential methylation sites (DMSs) were first called using Methy kit (R 3.3.2) (37) between P2 and P10 from the same individual with a false discovery rate lower than 0.05. DMSs were shared in at least two individuals in either control or LBN groups, and DMSs falling within 100 base pairs were then merged into DMRs.

### Calculation of DNA methylation level/percentage and δ methylation

The methylation percentage/level was calculated as the ratio of the methylated read counts over the sum of both methylated and unmethylated read counts for a single CpG site or across CpGs for a region. The δ methylation was calculated using the log2 transformation of the ratio of methylation level in the P10 sample and the methylation level in the P2 sample, defined as $\log_2(P10/P2)$. Increased methylation in P10 is shown as a positive value, whereas decreased methylation in P10 is shown as a negative value.

## PCA and k-means clustering

Before PCA analysis, DNA methylation level of DMRs is batch-corrected by using *removeBatchEffect* function from *limma* (R package) with setting cohorts as batches. PCA analysis was performed on these batch-corrected DMRs by using *IncrementalPCA* function from scikit-learn (38) using python 2 for both Figs 2 and 3. The value of k was set to 10 for the k-means clustering based on a preliminary hierarchical clustering analysis. A DNA methylation heat map was generated with heat map.2 function in R 3.5.0 and a δ methylation heat map was generated using Java TreeView (39).

## Gene analysis

Genes associated with DMRs were identified using Homer 4.7 (40). For subsequent analyses, genes were kept if (1) CpGs were located within 20 kb of TSS in intergenic, promoter-TSS, and TTS positions; (2) CpGs were located within gene exons or introns. Gene ontology analysis was performed using Metascape (41) using the hyper-geometric test with corrected *P*-value lower than 0.05.

## Data access

Reads and processed data from RRBS assays have been submitted to the Gene Expression Omnibus data repository (http://www.ncbi.nlm.nih.gov/geo/) under accession number GSE119640.

# Supplementary Information

# Acknowledgements

This work was supported in part by the grants from the National Institutes of Health (MH096889, MH096889-S1, and MH73136 to TZ Baram and A Mortazavi).

## Author Contributions

S Jiang: conceptualization, data curation, formal analysis, methodology, and writing—original draft, review, and editing.
N Kamei: conceptualization, data curation, formal analysis, methodology, and writing—original draft, review, and editing.
JL Bolton: investigation and methodology.
XY Ma: investigation and methodology.
HS Stern: investigation, methodology, and writing—review and editing.
TZ Baram: conceptualization, resources, data curation, formal analysis, supervision, funding acquisition, investigation, methodology, project administration, supervision, and writing—original draft, review, and editing.
A Mortazavi: conceptualization, data curation, formal analysis, supervision, funding acquisition, investigation, methodology, project administration, and writing—original draft, review, and editing.

## Conflict of Interest Statement

The authors declare that they have no conflict of interest.

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
