## [Reviewer comments · Life Science Alliance]

Life Science Alliance

Intra-individual methylomics detects the impact of early-life adversity

Shan Jiang, Noriko Kamei, Jessica Bolton, Xinyi Ma, Hal Stern, Tallie Baram, and Ali Mortazavi
DOI: <https://doi.org/10.26508/lsa.201800204>

Corresponding author(s): Ali Mortazavi, University of California, Irvine and Tallie Baram, UC Irvine

Review Timeline:

Submission Date:	2018-09-28
Editorial Decision:	2018-11-06
Revision Received:	2019-01-01
Editorial Decision:	2019-02-11
Revision Received:	2019-03-10
Accepted:	2019-03-11

Scientific Editor: Andrea Leibfried

Transaction Report:

November 6, 2018

Re: Life Science Alliance manuscript #LSA-2018-00204

Dr. Ali Mortazavi
University of California, Irvine
Developmental and Cell Biology
2300 Biological Sciences 3
Irvine, CA 92697

Dear Dr. Mortazavi,

Thank you for submitting your manuscript entitled "Intra-individual methylomics detects the impact of early-life adversity" to Life Science Alliance. The manuscript was assessed by expert reviewers, whose comments are appended to this letter.

As you will see, the reviewers appreciate the analyses performed and find it potentially important. However, they are currently not convinced that your conclusions are sufficiently supported, especially since it remains somewhat unclear how / how rigorously the analyses were performed. All reviewers provide constructive input on how to demonstrate robustness of the dataset and how to turn it into a valuable resource for others. Given this input, we would like to invite you to revise your work, addressing all points raised by the reviewers. We think it is not entirely clear whether the conclusions will still stand upon revision, and we would like to point out that we'll need strong support from the reviewers on the revised version to allow publication here.

The typical timeframe for revisions is three months. Please note that papers are generally considered through only one revision cycle.

Thank you for this interesting contribution to Life Science Alliance. We are looking forward to receiving your revised manuscript.

Sincerely,

- A letter addressing the reviewers' comments point by point.
- An editable version of the final text (.DOC or .DOCX) is needed for copyediting (no PDFs).
- High-resolution figure, supplementary figure and video files uploaded as individual files: See our detailed guidelines for preparing your production-ready images, <http://life-science-alliance.org/authorguide>
- Summary blurb (enter in submission system): A short text summarizing in a single sentence the study (max. 200 characters including spaces). This text is used in conjunction with the titles of papers, hence should be informative and complementary to the title and running title. It should describe the context and significance of the findings for a general readership; it should be written in the present tense and refer to the work in the third person. Author names should not be mentioned.

B. MANUSCRIPT ORGANIZATION AND FORMATTING:

Full guidelines are available on our Instructions for Authors page, <http://life-science-alliance.org/authorguide>

Reviewer #1 (Comments to the Authors (Required)):

The study of "Intra-individual methylomics detects the impact of early-life adversity" by Jiang, Kamei et al. addresses a relevant question in the field and the approach used (swab over blood drawn)

has the benefit to better translated into future human studies. The major limit of this study is that it fails to provide meaningful insights on the downstream impact of intra-individual differences in DNA methylation. The GO terms that authors presented in Figure 4 are quite generic, although relevant. The authors state two aims for their study (discussion on page 7) and their results generally meet both. However, a deeper analysis of the downstream impact of intra-individual differences in DNA methylation would greatly increase the impact of this study.

Major points

- Based on what the authors show in Figure 2, I suggest to remove "in conjunction with experience" from this sentence on page 5.
"These data demonstrate that development and age modify the buccal swab methylome (Reizel et al. 2018; Smith et al. 2015; Eipel et al. 2016; Horvath and Raj 2018) in conjunction with experience."
- Figure 2a, what are the blue and orange bars directly above the heatmaps? There are two heatmaps per group. Are these P2 and then P10, and as in 2b? Presumably so, but it should be clearly indicated for 2a, then it makes more sense if it carries through to 2b, not vice versa. Overall, the heatmaps don't clearly indicate DMRs, as most rows show little change.
- It's not clear how the authors can justify the following statement, as some member of different groups show the same delta methylation. "We then examined the intra-individual methylation changes in detail and found that the patterns of changes in methylation within an individual were distinct depending on group assignment (Figure 3A)". Both groups, Control and LBN show hyper and hypo methylation in each cluster. For example, clusters 1,2, 5, 8 -10, show both hyper and hypo DMRs for individuals in both groups. Can the authors point to truly distinct clusters? This is imperative as it is the heart of the manuscript. While 3b does a better job of separating groups based on experience with age, it hard to tell if this is a biased effect of the cloud drawn around the colored dots. How do the authors justify grouping in what looks like outliers? Presumably this is from the orientation of the PCA. Since PC1 explains 49%, would it make more sense to have this in the x-plane instead of the z-plane. The authors could a better job of explaining how PC3 (4.9%) best separates controls from LBN, when PC1 is at 49%. Furthermore, Figure S6 seems to be best illustration of differential methylation by group and should be moved to a main figure.
- Figure S5 shows no significant differences. What are the authors hoping to illustrate?
- Overall figure legends could be more informative. They are essentially just titles for each sub-figure?

Minor points

- LBN not described in main text.
- Fig 3B. PCA PC1 scale is not readable.
- Fig 4B-C. GO term plots. Too busy and somewhat confusing to track dots and labels. I suggested a different type of chart.
- Fig. S2. Suggest increasing size font of x-axis. Same for other similar plots in the supplement.
- Mapping efficiency is a bit low and the bisulfite conversion rate is not reported. Per se the number of shared CpGs and DMRs seem good, but I would want to see that the conversion rate is good as well.

Reviewer #2 (Comments to the Authors (Required)):

In this manuscript, the authors describe a DNA methylation signature of early adversity in buccal swab DNA from a rat model of simulated poverty. The authors describe a strong developmental change of DNA methylation over the first 10 days of life but also discern a distinct epigenetic signature of early life adversity in this peripheral tissue. Overall this is a conceptually highly interesting and relevant study that could guide detection of adversity signatures in humans.

There are, however, a number of methodological questions that remain open and strongly impact the potential impact of the findings.

- 1) More detail needs to be given on the methods for DMR detection. Initially, the authors describe that DMRs are defined for every pup between P2 and P10 - more detail needs to be given how robust this method is, as it uses an N of 1 with repeated measures. As I read it, the authors then check, how many DMRs are shared by 2 or more pups. Is the sharing more than expected by chance? This would be important to note. Also, when analyzing the DMRs in a group level analysis, would they survive FDR?
- 2) The authors report a cut-off of at least 10 reads per CpG to be included in the analysis. What was the average and range of coverage of the 3417 DMRs that are used for the main analysis? In figure S5, a histogram of change in DNA methylation (I guess between P2 and P10 - should be noted in figure legend) is presented. Most changes are well below 2%, so a coverage of at least 50 reads would be necessary to detect these with some - albeit borderline - confidence. More information on coverage of the DMRs in question would be helpful to assess how robust these differences could be discerned.
- 3) Two cohorts of animals are described in the figures, but not detail about cohorts is given in the methods. A brief reference to the stability of the findings across the two cohorts is made in the results and the reader is referred to Suppl Figures 2 and 3. However, in the figure legends no statistics for similarity are provided and Fig S3 would actually suggest quite strong cohort effects. Here more information of statistics is needed.
- 4) In the enrichment analyses (TF and pathways) it is not clear what background was used to test enrichment, if there is in fact significant enrichment or if the numbers reported are just descriptive.

Reviewer #3 (Comments to the Authors (Required)):

The study by Jiang et al examines the effects of early life adversity on DNA methylation in a rat model of early life adversity. Genome wide DNA methylation in buccal swabs was examined using RRBS at two-time points P2 and P10. DMR analysis reveals differences intra-individually and as a group between P2 and P10 but a principal component analysis reveals only separation by age but not by early life adversity experience. However, although absolute methylation levels didn't differ between the early life adversity groups, intra-individual changes in methylation between P2 and P10 (delta methylation) did differ between the groups. However absolute levels of methylation were not different. The genes that showed a difference in intraindividual difference between the adversity groups were mostly involved in transcriptional and developmental regulation. The authors propose that intraindividual differences in methylation at two-time points could serve as biomarkers of early life adversity.

Critique

This is an important and interesting analysis. There are several strengths for this paper. First, the

two time point measurements in the same living individuals allows for measuring the true impact of an intervention by measuring DNA methylation at base line and following the intervention in the same individual. This has not been done often to my knowledge. Second, the paper shows that buccal DNA is informative on age related as well as adversity related DNA methylation measurements, which has implications for human biomarker development. Third, the paper provides a method for assessing the impact of exposures and interventions on DNA methylation at multiple time points in living animals. Fourth, the intraindividual analysis offers a new insight into the personalized impact of early life adversity that might be lost when animals are grouped together. By comparing each individual to its own baseline interindividual differences including genetic and nongenetic confounders are excluded.

Comments

- a. A differential methylation analysis was done between P2 and P10 which was followed by PCA analysis This analysis couldn't discriminate between the early adversity group and controls. The authors should perform a differential methylation analysis between the adversity groups at P10 and show the results. If each individual is different before and after the adversity exposure as the authors show later (deltas are different), the groups should also be. We need to get an idea of how many differences are detected between the treatment groups, what is the scope of the difference and how many of the sites remain significant after adjustment?
- b. It will be nice to provide examples of the genes that are differentially methylated between ages, the real methylation values, not just significance, the size of the change and the standard deviation of the methylation values at each stage perhaps in a form of a chart.
- c. Are the genes that show intraindividual changes in response to adversity also genes that change with age in the control groups? Is there an interaction between age and adversity?
- d. The authors show intraindividual difference between P2 and P10 that is larger in the adversity group than in controls but there is no difference between the groups. How is this possible? Perhaps the baseline values are highly variable erasing the effects of the difference with adversity when they are averaged. We need to see the real methylation values for these genes at baseline and after adversity for each individual in the two groups to be able to assess this. Such a figure should be provided.
- e. The authors suggest that these CGs whose delta methylation between P10 and P2 is larger in the adverse group could serve as biomarkers for early life adversity. How do you measure delta and how do you define a predictive delta threshold in a person? In most cases you will want to know whether there was adversity in the early life history of a person. Baseline methylation is unknown. What exactly are you going to measure if there is a threshold that is derived as a group norm that the methylation measurement could compare to?

Reviewer #1 (Comments to the Authors (Required)):

The study of "Intra-individual methylomics detects the impact of early-life adversity" by Jiang, Kamei et al. addresses a relevant question in the field and the approach used (swab over blood drawn) has the benefit to better translated into future human studies. The major limit of this study is that it fails to provide meaningful insights on the downstream impact of intra-individual differences in DNA methylation. The GO terms that authors presented in Figure 4 are quite generic, although relevant. The authors state two aims for their study (discussion on page 7) and their results generally meet both. However, a deeper analysis of the downstream impact of intra-individual differences in DNA methylation would greatly increase the impact of this study.

We thank the Reviewer for appreciating the potential impact of our work. As suggested, we now provide—in designated sections--additional information and a discussion of the potential downstream impact of the differences in DNA methylation. In addition, rather than relying only on generic GO terms, we refine our categorization in the context of the age of the sample-producing animals and the cell type. The new information is included in Figure 4, in the text of the results and in the discussion.

Major points

(1) Based on what the authors show in Figure 2, I suggest to remove "in conjunction with experience" from this sentence on page 5. "These data demonstrate that development and age modify the buccal swab methylome (Reizel et al. 2018; Smith et al. 2015; Eipel et al. 2016; Horvath and Raj 2018) in conjunction with experience."

We removed the phrase, as requested.

(2) Figure 2a, what are the blue and orange bars directly above the heatmaps? There are two heatmaps per group. Are these P2 and then P10, and as in 2b ? Presumably so, but it should be clearly indicated for 2a, then it makes more sense if it carries through to 2b, not vice versa. Overall, the heatmaps don't clearly indicate DMRs, as most rows show little change.

We regret the missing labels in Figure 2A. Yes, the blue bar represents P2 and the red bar represents P10, same as in 2B. We added the labels in 2A, revised the figure significantly and enhanced the figure legend.

(3) It's not clear how the authors can justify the following statement, as some member of different groups show the same delta methylation. "We then examined the intra-individual methylation changes in detail and found that the patterns of changes in methylation within an individual were distinct depending on group assignment (Figure 3A)". Both groups, Control and LBN show hyper and hypo methylation in each cluster. For example, clusters 1,2, 5, 8 -10, show both hyper and hypo DMRs for individuals in both groups.

The Reviewer's point is well taken. Therefore, we modified the sentence to read: "We then examined the intra-individual methylation changes in detail and found features of the changes in methylation that were distinct depending on group assignment." We then proceeded to analyze these patterns in more detail, as requested.

Can the authors point to truly distinct clusters? This is imperative as it is the heart of the manuscript. While 3b does a better job of separating groups based on experience with age, it hard to tell if this is a biased effect of the cloud drawn around the colored dots. How do the authors justify grouping in what looks like outliers? Presumably this is from the orientation of the PCA. Since PC1 explains 49%, would it make more sense to have this in the x-plane instead of the z-plane. The authors could a better job of explaining how PC3 (4.9%) best separates controls from LBN, when PC1 is at 49%.

The Reviewer raises an interesting point, pertaining to the key 'drivers' of the differential methylation in control and adversity (LBN) groups. Both groups are developing rapidly from P2 to P10. In addition, the LBN experience modifies this development. These two parameters are reflected in the PCA. It seems that development dominates, contributing more than experience, likely reflected in PC1 whereas the adversity experience is reflected as a smaller 'driver' in PC4 (a cleaned re-analysis revealed PC4 as the discriminant component between controls and the LBN group).

We grouped the samples within the PCA by looking at the "0" line of PC4 (side view in Figure 3A and top view in Figure S8A), which separates individuals by DMRs with positive and negative weights. One of the control samples clustered with the LBNs and the rest were separated by the "0" line. In the same analysis, PC1 explained 49% of the variance, representing the difference among individuals having more DMRs with higher methylation in P10 or in P2.

Furthermore, Figure S6 seems to be the best illustration of differential methylation by group and should be moved to a main figure.

We appreciate the recommendation. The original Figure S6 is now Figure 3B and 3C of the main manuscript.

(4) Figure S5 shows no significant differences. What are the authors hoping to illustrate? We apologize for the ambiguity. This figure demonstrates that there are no technical artifacts or single individuals that are driving the significant difference between control and LBN or cohort 1 and cohort 2. Specifically, there are no shifting of the distribution profiles in control, LBN, cohort 1 and cohort 2.

(5) Overall figure legends could be more informative. They are essentially just titles for each sub-figure?

As requested, we have now updated and enhanced the legends for both the main and the supplementary figures to make them more informative.

Minor points

(6) LBN not described in main text.

We describe LBN in the Introduction, noting that "Specifically, we imposed 'simulated poverty' by raising pups for a week (from postnatal day P2 to P10) in cages with limited bedding and nesting materials (LBN). This manipulation disrupts the care provided by the rat dam to her pups and results in profound yet transient stress in the pups, devoid of major weight-loss or physical changes. This transient experience provokes significant and life-long deficits in memory and generates increases in emotional measures of anhedonia and depression (Ivy et al. 2010; Bolton et al. 2018; Lister et al. 2013)."

(7) Fig 3B. PCA PC1 scale is not readable

In the revised paper, we altered the sequence of panels. As requested, we augmented and improved the visibility of the PC1 scale

(8) Fig 4B-C. GO term plots. Too busy and somewhat confusing to track dots and labels. I suggested a different type of chart.

As suggested by the Reviewer, we revamped the original Figure 4B,C. In the revised figure, we focus on the genes that contribute most to the differences between LBN and controls. These are highlighted in the new Figure 4B and D.

(9) Fig. S2. Suggest increasing size font of x-axis. Same for other similar plots in the supplement. We increased the font sizes of all of the axes in both main and supplementary figures.

(10) Mapping efficiency is a bit low and the bisulfite conversion rate is not reported. Per se the number of shared CpGs and DMRs seem good, but I would want to see that the conversion rate is good as well.

We now provide the ratio of methylated C in the CpG islands for each sample in the new supplementary Figure S1.

Reviewer #2 (Comments to the Authors (Required)):

In this manuscript, the authors describe a DNA methylation signature of early adversity in buccal swab DNA from a rat model of simulated poverty. The authors describe a strong developmental change of DNA methylation over the first 10 days of life but also discern a distinct epigenetic signature of early life adversity in this peripheral tissue. Overall this is a conceptually highly interesting and relevant study that could guide detection of adversity signatures in humans. We truly thank the Reviewer for appreciating the significance and potential impact of our paper.

There are, however, a number of methodological questions that remain open and strongly impact the potential impact of the findings.

1) More detail needs to be given on the methods for DMR detection. Initially, the authors describe that DMRs are defined for every pup between P2 and P10 - more detail needs to be given how robust this method is, as it uses an N of 1 with repeated measures. As I read it, the authors then check, how many DMRs are shared by 2 or more pups. Is the sharing more than expected by chance? This would be important to note. Also, when analyzing the DMRs in a group level analysis, would they survive FDR?

As suggested by the Reviewer, we updated the methods for DMR detection. The Reviewer is correct, we detected differential methylation sites (DMSs) with an $FDR < 0.05$ for each individual and only considered DMSs shared by 2 or more pups in the same group (for controls, the significance of this sharing is $p = 2.1 \times 10^{-10}$, chi-squared test adjusted with Fisher's exact test; for LBNs, the significance is $p = 1.7 \times 10^{-14}$). We then merged DMSs falling within 100 basepairs (bp) to get 3417 DMRs for downstream analysis. We detected the DMRs at the group level between P2 and P10 and found that 3060 DMRs (89.5% of 3417 DMRs) survived $FDR < 0.05$.

2) The authors report a cut-off of at least 10 reads per CpG to be included in the analysis. What was the average and range of coverage of the 3417 DMRs that are used for the main analysis? In figure S5, a histogram of change in DNA methylation (I guess between P2 and P10 - should be noted in figure legend) is presented. Most changes are well below 2%, so a coverage of at least 50 reads would be necessary to detect these with some - albeit borderline - confidence. More information on coverage of the DMRs in question would be helpful to assess how robust these differences could be discerned.

We regret the ambiguity of the original description of our calculation of delta methylation. The average of the coverage of the 3417 DMRs ranged between 32-63 reads across individuals. We calculated delta methylation by using fold changes between P10 and P2 at each DMRs rather than the differences ($P10 - P2$) and the x-axis in original figure S5 (now part of Figure S6) are \log_2 scaled. For example, if one DMR is 100% methylated at P10 but 60% methylated at P2, their difference will be 40% (100%-60%) but their \log_2 fold change in our manuscript will be only 0.74 ($\log_2(100\%/60\%)$). This is the source of the fact that most of the changes in the figures are around 2%.

3) Two cohorts of animals are described in the figures, but not detail about cohorts is given in the methods. A brief reference to the stability of the findings across the two cohorts is made in the results and the reader is referred to Suppl Figures 2 and 3. However, in the figure legends not statistics for

similarity are provided and Fig S3 would actually suggest quite strong cohort effects. Here more information of statistics is needed.

We appreciate the Reviewer's comments about the cohort differences. In the revised methods we provide information about these cohorts- they differed only in the date of the experiment. As shown in our Figure S4 and S5 (updated version of original Figure S3 and S2), we do observe a significant cohort difference on methylation distribution profiles ($p < 2.2 \times 10^{-16}$, Mann-Whitney U Test, Figure S5A) before batch correction. However, the cohort effects disappear ($p = 0.2197$, Mann-Whitney U Test, Figure S5D) after batch and covariate effects are removed by using *limma* (Methods). Similarly, minimal cohort effects are observed in the PCA analysis by comparing before (Figure S4C) and after batch correction (Figure S4D). We updated the figure legends in Figure S5 to state more explicitly our statistical approaches to the cohort effects.

4) In the enrichment analyses (TF and pathways) it is not clear what background was used to test enrichment, if there is in fact significant enrichment or if the numbers reported are just descriptive.

In the original version, the analyses were done by using rat ensemble ID as input and human as background to test enrichment. GO terms with $FDR < 0.05$ were reported in the figure. However the traditional GO analyses did not account for the fact that a relatively few DMRs contributed maximally to the PCA-derived prediction of belonging to a given group. Therefore, in the revision, we focused instead on the genes that contributed most to the differences between LBN and controls. These are highlighted in the new figure 4B and D. We also updated the GO terms by using rat ensemble ID and rat as background to test the enrichment for age-related DMRs. These terms are shown in the new supplemental Figures S2 and S3.

Reviewer #3 (Comments to the Authors (Required)):

The study by Jiang et al examines the effects of early life adversity on DNA methylation in a rat model of early life adversity. Genome wide DNA methylation in buccal swabs was examined using RRBS at two-time points P2 and P10. DMR analysis reveals differences intra-individually and as a group between P2 and P10 but a principal component analysis reveals only separation by age but not by early life adversity experience. However, although absolute methylation levels didn't differ between the early life adversity groups, intra-individual changes in methylation between P2 and P10 (delta methylation) did differ between the groups. However absolute levels of methylation were not different. The genes that showed a difference in intraindividual difference between the adversity groups were mostly involved in transcriptional and developmental regulation. The authors propose that intraindividual differences in methylation at two-time points could serve as biomarkers of early life adversity.

Critique

This is an important and interesting analysis. There are several strengths for this paper. First, the two time point measurements in the same living individuals allows for measuring the true impact of an intervention by measuring DNA methylation at base line and following the intervention in the same individual. This has not been done often to my knowledge. Second, the paper shows that buccal DNA is informative on age related as well as adversity related DNA methylation measurements, which has implications for human biomarker development. Third, the paper provides a method for assessing the impact of exposures and interventions on DNA methylation at multiple time points in living animals. Fourth, the intraindividual analysis offers a new insight into the personalized impact of early life adversity that might be lost when animals are grouped together. By comparing each individual to its own baseline interindividual differences including genetic and nongenetic confounders are excluded.

We are grateful to the reviewer for appreciating the strengths and implications of our work.

Comments

(1) A differential methylation analysis was done between P2 and P10 which was followed by PCA analysis. This analysis couldn't discriminate between the early adversity group and controls. The authors should perform a differential methylation analysis between the adversity groups at P10 and show the results. If each individual is different before and after the adversity exposure as the authors show later (deltas are different), the groups should also be. We need to get an idea of how many differences are detected between the treatment groups, what is the scope of the difference and how many of the sites remain significant after adjustment?

As requested by the Reviewer (as well as by Reviewer 2), we have now performed a group level analysis, comparing DNA methylation in P10 rats that had been reared in adversity to those reared in a typical environment. We found 2152 DMRs (62.9% of 3417 DMRs) survived FDR<0.05. These DMRs could separate P10 LBNs from P10 controls by PC4 with about 5% variance explained.

(2) It will be nice to provide examples of the genes that are differentially methylated between ages, the real methylation values, not just significance, the size of the change and the standard deviation of the methylation values at each stage perhaps in a form of a chart.

We added two supplementary figures, S2 and S3, to show differentially methylated DMRs between ages. We provide the separation between P2 and P10 individuals by PC2 when considering both control and LBNs (Figure S2) or only controls (Figure S3). We also demonstrate the absolute methylation profiles in both figures to show that the most weighted DMRs in PC2 represent differential methylation level between P10 and P2. We added a supplementary Table S2 to present the corresponding methylation values and the average size of change across all samples.

(3) Are the genes that show intra individual changes in response to adversity also genes that change with age in the control groups? Is there an interaction between age and adversity?

This is a complex question to address especially because the levels of analysis in this series of experiments is methylation on specific DMRs rather than at the levels of genes. Thus, there are thousands (3417) sites that contribute to some degree to separations of individual rats by age and/or by experience. To estimate the DMRs (and genes) that contribute meaningfully to age group or experience group, we examined the DMRs and genes that have the most weight in the PCs that contribute to group prediction. Thus, we can compare the genes illustrated in Figure 4A to those depicted in Figure 4C. Inspection of these lists (as well as of the gene categories in 4B and 4D) reveals relatively little overlap. In summary, whereas it is likely that gene methylation is influenced by both age and experience, our data suggest that the genes contributing maximally to the LBN-specific methylation changes are distinct from those contributing maximally to the age difference between P2 and P10.

(4) The authors show intraindividual difference between P2 and P10 that is larger in the adversity group than in controls but there is no difference between the groups.

How is this possible? Perhaps the baseline values are highly variable erasing the effects of the difference with adversity when they are averaged. We need to see the real methylation values for these genes at baseline and after adversity for each individual in the two groups to be able to assess this. Such a figure should be provided.

The Reviewer's point is well taken, and we now address the confusion. On P2, prior to group assignment, all pups are identical. Thus, there is not difference in methylation levels of pups that are subsequently assigned to either control or adversity groups. Second, the revised Figure 3 now shows that the delta methylation changes are not simply larger or smaller in the LBN vs the control group: they go often in opposite directions. Thus, the middle bars in Figure 3B illustrate the average as well as the directionality of delta methylation in the LBN vs. controls, focusing on the DMRs contributing maximally to the group differences. As apparent, directionality as well as magnitude of delta methylation distinguish the groups.

(5) The authors suggest that these CGs whose delta methylation between P10 and P2 is larger in the adverse group could serve as biomarkers for early life adversity. How do you measure delta and how do you define a predictive delta threshold in a person?

The Reviewer's point is well taken. In the new Figure 4D we provide a list of genes that might be implicated in adversity in an immature rat. We do not know if the same epigenomic signature will be observed in human infants. To that end, we are assessing the degree of adversity experienced by a human cohort in parallel to sampling DNA. We hope to be able to address the Reviewer's cogent question in future publications.

In most cases you will want to know whether there was adversity in the early life history of a person. Baseline methylation is unknown. What exactly are you going to measure if there is a threshold that is derived as a group norm that the methylation measurement could compare to?

This query is valid. As mentioned above, in the pilot human study we are sampling each baby twice. Our hope is that, should our results be translationally and clinically useful, one of two scenarios will obtain. First high-risk neonates may routinely contribute a swab upon hospital discharge for comparison purposes. Alternatively, we will obtain neonatal DNA from stored samples collected routinely for genetic screens.

February 11, 2019

RE: Life Science Alliance Manuscript #LSA-2018-00204-TR

Dr. Ali Mortazavi
University of California, Irvine
Developmental and Cell Biology
2300 Biological Sciences 3
Irvine, CA 92697

Dear Dr. Mortazavi,

Thank you for submitting your revised manuscript entitled "Intra-individual methylomics detects the impact of early-life adversity". As you will see, the reviewers appreciate the introduced changes and we would thus be happy to publish your paper in Life Science Alliance pending final revisions necessary to meet our formatting guidelines:

- the number of animals used is currently hidden in the cohort tables, please make n used more visible (eg in figure legends or in Material and Methods section).
- Figure panels S3C and S8A are mentioned in the manuscript text, while the other panels of these figures are not. Please be consistent and either call out all panels or none.

A. FINAL FILES:

-- High-resolution figure, supplementary figure and video files uploaded as individual files: See our detailed guidelines for preparing your production-ready images, <http://life-science-alliance.org/authorguide>

-- Summary blurb (enter in submission system): A short text summarizing in a single sentence the study (max. 200 characters including spaces). This text is used in conjunction with the titles of papers, hence should be informative and complementary to the title. It should describe the context and significance of the findings for a general readership; it should be written in the present tense

and refer to the work in the third person. Author names should not be mentioned.

B. MANUSCRIPT ORGANIZATION AND FORMATTING:

Full guidelines are available on our Instructions for Authors page, <http://life-science-alliance.org/authorguide>

Sincerely,

Reviewer #1 (Comments to the Authors (Required)):

In the revised version of the manuscript entitled "Intra-individual methylomics detects the impact of

early-life adversity" by Jiang et al., the authors have made substantial improvements to the manuscript and study as a whole. They provided a detailed response to reviewers comments, updating text and figures as suggested. I have no further concerns.

Reviewer #2 (Comments to the Authors (Required)):

The authors have now addressed all remaining concerns and clarified the methods. An important paper.

March 11, 2019

RE: Life Science Alliance Manuscript #LSA-2018-00204-TRR

Dr. Ali Mortazavi
University of California, Irvine
Developmental and Cell Biology
2300 Biological Sciences 3
Irvine, CA 92697

Dear Dr. Mortazavi,

Thank you for submitting your Research Article entitled "Intra-individual methylomics detects the impact of early-life adversity". It is a pleasure to let you know that your manuscript is now accepted for publication in Life Science Alliance. Congratulations on this interesting work.

DISTRIBUTION OF MATERIALS:

Again, congratulations on a very nice paper. I hope you found the review process to be constructive and are pleased with how the manuscript was handled editorially. We look forward to future exciting submissions from your lab.

Sincerely,
